# Integrative Metabolomic Analysis of Serum and Selected Serum Exosomal microRNA in Metastatic Castration-Resistant Prostate Cancer

**DOI:** 10.3390/ijms25052630

**Published:** 2024-02-23

**Authors:** Daniel Evin, Andrea Evinová, Eva Baranovičová, Miroslava Šarlinová, Jana Jurečeková, Peter Kaplán, Hubert Poláček, Erika Halašová, Róbert Dušenka, Lukáš Briš, Martina Knoško Brožová, Monika Kmeťová Sivoňová

**Affiliations:** 1Department of Medical Biochemistry, Jessenius Faculty of Medicine in Martin, Comenius University in Bratislava, 03601 Martin, Slovakia; daniel.evin@unm.sk (D.E.); jana.jurecekova@uniba.sk (J.J.); peter.kaplan@uniba.sk (P.K.); brozova8@uniba.sk (M.K.B.); 2Clinic of Nuclear Medicine, Jessenius Faculty of Medicine in Martin, University Hospital in Martin, Comenius University in Bratislava, 03601 Martin, Slovakia; hubert.polacek@uniba.sk; 3Biomedical Center Martin, Jessenius Faculty of Medicine in Martin, Comenius University in Bratislava, 03601 Martin, Slovakia; andrea.evinova@uniba.sk (A.E.); eva.baranovicova@uniba.sk (E.B.); miroslava.sarlinova@uniba.sk (M.Š.); erika.halasova@uniba.sk (E.H.); 4Clinic of Urology, Jessenius Faculty of Medicine in Martin, University Hospital in Martin, Comenius University in Bratislava, 03601 Martin, Slovakia; robert.dusenka@uniba.sk (R.D.); lukas.bris@unm.sk (L.B.)

**Keywords:** metastatic castration-resistant prostate cancer, microRNA, ^1^H-NMR metabolomics

## Abstract

Metastatic castration-resistant prostate cancer (mCRPC) remains a lethal disease due to the absence of effective therapies. A more comprehensive understanding of molecular events, encompassing the dysregulation of microRNAs (miRs) and metabolic reprogramming, holds the potential to unveil precise mechanisms underlying mCRPC. This study aims to assess the expression of selected serum exosomal miRs (miR-15a, miR-16, miR-19a-3p, miR-21, and miR-141a-3p) alongside serum metabolomic profiling and their correlation in patients with mCRPC and benign prostate hyperplasia (BPH). Blood serum samples from mCRPC patients (*n* = 51) and BPH patients (*n* = 48) underwent metabolome analysis through ^1^H-NMR spectroscopy. The expression levels of serum exosomal miRs in mCRPC and BPH patients were evaluated using a quantitative real-time polymerase chain reaction (qRT-PCR). The ^1^H-NMR metabolomics analysis revealed significant alterations in lactate, acetate, citrate, 3-hydroxybutyrate, and branched-chain amino acids (BCAAs, including valine, leucine, and isoleucine) in mCRPC patients compared to BPH patients. MiR-15a, miR-16, miR-19a-3p, and miR-21 exhibited a downregulation of more than twofold in the mCRPC group. Significant correlations were predominantly observed between lactate, citrate, acetate, and miR-15a, miR-16, miR-19a-3p, and miR-21. The importance of integrating metabolome analysis of serum with selected serum exosomal miRs in mCRPC patients has been confirmed, suggesting their potential utility for distinguishing of mCRPC from BPH.

## 1. Introduction

Prostate cancer (PC) stands as the second most prevalent cancer globally, exhibiting the highest incidence rates in North and South America, Europe, Australia, and the Caribbean [1]. Despite being commonplace, the existing diagnostic methods for PC, incorporating a digital rectal examination (DRE), serum prostate-specific antigen (PSA) levels, and a transrectal ultrasound (TRUS)-guided biopsy, remain unsatisfactory [2]. Prostate-specific antigen density (PSAD) may serve as a more reliable predictor of clinically significant PC compared to PSA alone. The PSAD should be considered in evaluating patients at risk of PC who may require additional testing or a biopsy. This parameter can help differentiate elevated PSA levels due to benign conditions from those indicative of PC, potentially improving diagnostic accuracy and patient care [3]. The necessity for more specific biomarkers in PC detection is evident due to the limitations of the PSA test, which lacks specificity and may lead to unnecessary biopsies. The continual progress in urinary biomarker research, including the development of RNA-based tests and the exploration of exosomes as carriers of PC-specific molecules, holds promise for improving the specificity of PC detection [4].

Radiological imaging techniques, such as multiparametric magnetic resonance imaging (mpMRI), and nuclear medicine methods, notably skeletal scintigraphy and positron emission tomography (PET) utilizing ^68^Ga prostate-specific membrane antigen (PSMA) ligands, play pivotal roles in diagnosing the advanced stages of PC [5]. There is also an increasing focus on the MRI-guided prostate biopsy [6]. Metastatic castration-resistant prostate cancer (mCRPC) represents the terminal stage of PC, characterized by the failure of antiandrogen therapy and distant metastases predominantly to the skeleton [7]. This condition is inevitably linked with a grim prognosis for the patient [8]. Recent advances in the systemic treatment of mCRPC encompass novel androgen receptor-targeted agents, chemotherapy with Docetaxel and Cabazitaxel, immunotherapy, and nuclear medicine therapies such as ^223^RaCl_2_ and ^177^Lu prostate-specific membrane antigen (PSMA) [9].

Cancer cells have long been recognized for their extensive metabolic alterations, and the reprogramming of the cellular energy metabolism represents an emerging hallmark of cancer, exemplified by the Warburg effect [10]. In normal prostate epithelial cells, aerobic conditions typically lead to glycolysis instead of oxidative metabolism. These cells utilize glucose and aspartate to synthesize and secrete citrate into the lumen, a crucial component of semen [11,12]. This metabolic profile is a consequence of zinc accumulation in prostate cells, inhibiting the tricarboxylic acid (TCA) cycle enzyme, m-aconitase [13]. The progression of PC involves a decrease in zinc concentrations, leading to the reactivation of m-aconitase and the initiation of citrate oxidation through the TCA cycle [14]. In contrast to many tissues in the human body, the metabolism of primary PC cells is characterized by high lipogenesis, lower glycolysis, and dependence on oxidative phosphorylation [15]. Consequently, citrate can be exported to the cytoplasm and converted back into acetyl-CoA for de novo synthesis of fatty acids and cholesterol [16]. Characteristic of advanced metastatic stages, PC cells become highly glycolytic, inhibiting mitochondrial respiration and exhibiting the Warburg effect. Additionally, the deregulated anabolism/catabolism of fatty acids and amino acids has been identified as a metabolic regulator supporting cancer cell growth [17].

MiRs are small non-coding RNA molecules that regulate gene expression and can be dysregulated in various types of diseases, including PC [18]. Analyzing their expression in prostatic tissues or biological fluids, such as blood, can help identify specific patterns associated with the disease. Additionally, these panels provide prognostic markers, with specific miRs linked to tumor aggressiveness and prognosis, offering insights into disease spread and treatment response. Furthermore, changes in miRs expression during treatment can be monitored for insights into therapy effectiveness and potential therapeutic targets in manipulating miRs for influencing PC growth or metastasis. The amalgamation of miRs and PSA data exhibited heightened sensitivity and specificity for PC diagnosis compared to utilizing PSA alone. This integrated diagnostic strategy outperformed the use of PSA in isolation, where sensitivity and specificity are increased [19].

The analysis of a miRs panel in PC offers additional insights into the molecular aspects of the disease. The presence of malignancy in an organism is evident through an altered metabolism, and it is known that many miRs play a critical role in regulating cellular metabolism under normal and pathological conditions. Therefore, the present study aimed to determine the changes in the expression of selected exosomal miRs (miR-15a, miR-16, miR-19a-3p, miR-21, and miR-141a-3p) together with the serum metabolomic profiles between mCRPC and BPH patients, for the discrimination of mCRPC from benign prostate disease.

## 2. Results

### 2.1. Relative Changes in Blood Metabolites

Statistically significant differences were observed in the relative concentrations of blood serum metabolites associated with energy metabolism, including lactate, acetate, citrate, 3-hydroxybutyrate, and the essential amino acids BCAAs between mCRPC patients and the BPH group (Table 1). The most notable alterations were evident for citrate (relative difference of 38%) and acetate (relative difference of 24%). Valine and leucine, sharing numerous biochemical pathways, exhibited very similar changes, both decreasing by about 18% in the mCRPC group compared to the BPH group. The alteration in blood serum levels of their third co-metabolite isoleucine was weaker, with a marginally significant decrease of 11% in mCRPC patients against BPH patients. Figure 1 presents boxplots illustrating the significantly altered metabolites in the serum of mCRPC patients and the BPH group.

A comparison of median serum concentrations of metabolites such as alanine, glucose, pyruvate, phenylalanine, tyrosine, glutamine, threonine, lysine, branched-chain keto acids (BCKAs), creatine, creatinine, proline, histidine, succinate, and tryptophan did not reveal significant differences.

### 2.2. Exosomal miRNA Expression in the Serum Samples of mCRPC

Table 2 presents the fold regulation levels of selected exosomal miRs, namely miR-15a, miR-16, miR-19a-3p, miR-21, and miR-141a-3p. The results demonstrate that the expression levels of serum exosomal miR-15a, miR-16, miR-19a-3p, and miR-21 were significantly down-regulated in mCRPC patients compared with BPH patients (*p* < 0.05). Among them, the highest threefold decrease was observed in the expression level of miR-16 in mCRPC compared to BPH patients. However, the expression level of miR-141a-3p did not exhibit a significant variation within the mCRPC patient group compared to the BPH group (*p* > 0.05).

### 2.3. Correlations between miRs and Metabolites Levels

We investigated the correlation between the expression profile of selected serum exosomal miRs and metabolites in mCRPC patients (Table 3). Lactate showed a significant correlation with miR-15a, miR-16, and miR-19a-3p (*p* < 0.05), respectively. The results demonstrated that citrate and lysine were negatively correlated with serum miR-15a, miR-19a-3p, and miR-21 (*p* < 0.05), respectively. A significant correlation of 3-hydroxybutyrate with miR-15a and miR-21 (*p* < 0.05) was also observed. Furthermore, acetate was positively correlated with miR-19a-3p, miR-21, and miR-141a-3p (*p* < 0.05), respectively. Additionally, glutamine levels were positively correlated with miR-15a and miR-16 (*p* < 0.05). BCAAs did not show a significant correlation with selected exosomal miRs.

## 3. Discussion

As of today, mCRPC remains an incurable disease, and ongoing research is focused on novel therapeutic agents aimed at maximizing the survival and quality of life for these patients. Consequently, it is crucial to identify critical events in the progression of prostate malignancy through the integration of metabolomic and other omics techniques. NMR spectroscopy has proven to be a suitable methodology for evaluating the metabolic profile, effectively distinguishing between mCRPC patients and those with benign prostatic hyperplasia. Our findings indicate that certain metabolites, such as lactate, acetate, and citrate, are present in higher concentrations in the blood serum of mCRPC patients, while others, including 3-hydroxybutyrate and BCAAs, are decreased. Furthermore, our investigation reveals that serum exosomal miRs, specifically miR-15a, miR-16, miR-19a-3p, and miR-21, when combined with lactate, citrate, and acetate, exhibit significant potential for the discernment of BPH from mCRPC.

Normal prostate epithelial cells exhibit an inefficient energy metabolism characterized by the inactivation of the TCA cycle, resulting in high citrate production [11]. The neoplastic transformation in prostate cells coincides with the restoration of the TCA cycle and an increased generation of ATP from glucose oxidation [20]. In advanced stages of the disease, PC cells display an overexpression of glucose transporters and key glycolytic enzymes [11], leading to increased glucose consumption and lactate release [21]. The elevated lactate produced in hypoxic tumor areas is secreted into the extracellular environment [22], resulting in a high concentration of serum lactate (approximately 40 mM) observed in the serum of various cancer patients compared to the lactate concentration in healthy tissue and serum (1.5 to 3 mM) [23]. We observed elevated serum lactate levels in mCRPC patients compared to the BPH group, which aligns with previously reported findings in PC tissues [24,25,26]. We propose that cells in the tumor microenvironment produce lactate, serving as the source of increased blood serum lactate in patients with PC, and it may contribute to tumor cell invasion, metastasis, and immunosuppression [23].

Previously, it was demonstrated that intracellular citrate concentrations in the normal prostate peripheral zone exceed those in other soft tissues (10,000–15,000 nmol/g vs. 250–450 nmol/g). Similarly, citrate concentrations in prostatic ductal fluids are higher than in blood plasma (40–150 mM vs. approximately 0.2 mM) [27,28]. Numerous studies employing animal models, cell lines, and tissue extracts have consistently shown reduced citrate levels in the prostatic tissue of individuals with prostate adenocarcinoma compared to those with a normal prostate peripheral zone and BPH [29,30,31,32]. The exact cause of this reduction, whether due to altered citrate production (e.g., low Zn^2+^ levels leading to increased m-aconitase activity) and/or changes in citrate transportation, remains unclear [33].

In our study, we observed significantly higher blood serum citrate levels in mCRPC patients compared to BPH patients. We hypothesize that PC cells take up this serum citrate through specific transporters expressed in the plasma membrane [34,35], and intracellularly utilize it to support PC metabolism, proliferation, fatty acid synthesis, and protein acetylation [36]. Similar findings were reported by Buszewska-Forajta et al. (2022), who observed higher serum concentrations of citrate in a PC group compared to a BPH group, with no significant changes in citrate concentration based on the clinical stage of the tumor [37]. In contrast, Kumar et al. (2016) reported a significant decrease in citrate levels in filtered serum obtained from PC patients compared to BPH patients [32]. Huang et al. (2017) found a lower risk of T4 PC in men with higher serum citrate and fumarate concentrations compared to the controls, as determined by ultra-high performance liquid chromatography/mass spectroscopy (LC-MS) and gas chromatography/mass spectroscopy (GC-MS) [38].

PC bone metastases represent the final stage of the metastasis, associated with aggressive tumor growth and the development of primarily osteoblastic bone disease [39]. Typically, bone metastases in PC are osteoblastic, involving the deposition of newly formed bone, but they can also manifest as osteolytic, characterized by the destruction of normal bone, or mixed [40]. During bone formation, osteoblasts synthesize citrate, which becomes incorporated into the new bone. Conversely, during bone resorption, citrate is released from the bone into the blood [29]. Hence, we suggest that this constitutes another significant source of citrate in mCRPC patients, contributing to the elevated citrate levels observed in the serum.

Moreover, the process of oncogenesis is linked to alterations in the uptake and metabolism of amino acids. Amino acids serve as the building blocks of proteins and also act as intermediate metabolites fueling various biosynthetic pathways [41]. BCAAs, including leucine, isoleucine, and valine, are preferentially taken up by tumors. Because they are essential amino acids, their plasma levels are contingent on dietary intake and whole-body protein turnover [42,43]. Numerous studies have identified associations between circulating BCAAs levels and various human cancer types [44,45,46]. The study by Giskeødegård et al. (2015) demonstrated increased serum levels of BCAAs in PC patients compared to the BPH group [47]. Our findings, indicating significantly lower levels of serum BCAAs in the mCRPC group compared to the BPH group, align with the results of a few studies [48,49]. Similarly, Zhang et al. (2022) also observed significantly decreased serum BCAAs levels in PC patients with bone metastasis compared to PC or BPH patients, suggesting that downregulated BCAAs may be closely related to bone metastasis in PC progression [50]. We hypothesize that the decrease in serum BCAAs levels could result from higher BCAAs uptake through the L-type amino acid transporter LAT1 (SLC7A5), which is highly expressed in prostate tumor tissues [51], and the subsequent catabolism of BCAAs for energy production by PC cells. Furthermore, BCAAs metabolism may be affected by genetic mutations, the tumor microenvironment, food intake, and the individual’s health status [52].

Interestingly, we did not observe any differences in the corresponding BCKAs, such as ketoleucine, ketoisoleucine, and ketovaline, as well as glucose serum levels between the mCRPC and BPH groups. Additionally, we did not notice changes in alanine and glutamine levels, the main amino acids responsible for ammonia detoxification in extrahepatic tissues. Similarly, no differences were detected for other evaluated essential amino acids—phenylalanine, histidine, threonine, and tryptophan. These observations, coupled with the above-discussed facts, support the hypothesis of the selective utilization of BCAAs by PC cells as an energy substrate rather than their accelerated usage in proteosynthesis during the formation of new cells.

Another metabolite that we found altered in the serum of mCRPC patients relative to BPH patients is 3-hydroxybutyrate, a representative ketone body. Ketone body metabolism is dysregulated in various types of cancer, and most tumor cells are unable to use ketone bodies for energy due to abnormalities in the mitochondrial structure or function [53,54]. It has been reported that tumor cells can use ketone bodies as precursors for lipid synthesis rather than as energy substrates [55]. The study by Rodrigues et al. (2017) showed that the administration of 3-hydroxybutyrate may accelerate tumor growth [56]. However, there are also many studies presenting the anti-cancer effect of a ketogenic diet, a condition linked with increased ketone bodies levels [57].

Interestingly, Saraon et al. (2013) identified the ketogenic pathway as a novel bioenergetic pathway potentially involved in the progression of PC from a low-grade to a high-grade disease, followed by androgen independence [58]. Moreover, increased expression of both ketogenic and ketolytic enzymes (acetyl-CoA acetyltransferase 1, ACAT1; 3-hydroxybutyrate dehydrogenase 1, BDH1; 3-hydroxymethyl-3-methylglutaryl-CoA lyase, HMGCL; and 3-oxoacid CoA-transferase 1, OXCT1) was reported with PC progression, gradually increasing with the tumor grade [58,59]. Huang et al. (2017) showed that serum 3-hydroxybutyrate was associated with an increased risk of fatal PC in men diagnosed with metastatic disease [38].

Acetate, a short-chain fatty acid, serves as a substrate for the synthesis of acetyl coenzyme A (acetyl-CoA), primarily utilized in the de novo synthesis of fatty acids in PC cells, despite the abundance of circulating fatty acids. This metabolic phenotype is associated with PC progression and androgen independence [60,61]. Our findings align with previous studies reporting elevated serum acetate levels in various cancer types, including squamous oral carcinoma [62], lung cancer [63], and colon cancer [64]. We surmise that higher serum acetate levels may originate from the diet, with a significant portion generated by the metabolism of intestinal contents by the gut microbiome and/or from endogenous sources [65]. This pool of acetate serves as an alternative carbon source for fatty acid synthesis in cancer cells, potentially supporting the growth or metastasis of prostate tumors.

Numerous miRs have been reported to play significant roles in physiological and pathological processes, including cancer. Exosomes containing miRs, secreted by cancer cells, can be internalized by neighboring or distant recipient cells, facilitating tumor development [18]. The miRs from the miR-15/16 cluster are acknowledged as tumor suppressors, with documented reductions in various cancers, such as chronic lymphocytic lymphoma [66], pituitary adenomas [67], and PC [68]. Consistent with prior research, we observed a significant downregulation of serum exosomal miR-15a and miR-16 (fold regulation of −2.00 and −3.24, respectively) in mCRPC patients. Jin et al. (2018) proposed that miR-15a/16 inhibit components of the transforming growth factor-β (TGF-β) signaling pathways in the LNCaP cell line, implying a potential association with PC progression and metastasis [69]. Specifically, miR-15a and miR-16-1 exert their effects by targeting multiple oncogenes, including B-cell leukemia/lymphoma 2 (BCL2), myeloid cell leukemia-1 (MCL1), cyclin D1 (CCND1), and the wingless-type MMTV integration site family, member 3A (WNT3A). Moreover, the reduced expression of miR-15 and miR-16 in cancer-associated fibroblasts significantly enhances tumor growth and progression [70]. Additionally, miR-15a and miR-16-1 impact fatty acid metabolism, primarily downregulating fatty acid synthase (FASN) expression in mammary cells [71]. Thus, we speculate that the modulation of de novo fatty acid synthesis by miR-15a and miR-16, in conjunction with elevated serum levels of acetate and citrate, could potentially increase lipid synthesis in PC cells, and promote tumor invasiveness and metastatic ability.

Moreover, our study demonstrates that miR-15a and miR-16 expression correlates with serum lactate levels, potentially impacting the gene expression of key glycolytic enzymes, such as lactate dehydrogenase-A (LDH-A). Previous research has indicated that miR-16-5p directly targets LDH-A by binding to complementary regions in its 3′-UTR, resulting in the degradation and consequent reduction of lactate accumulation in non-small cell lung cancer (NSCLC) [72]. Additionally, we observed a significant correlation between miR-15a and miR-16 expression and serum glutamine levels, as lactate can also be generated from glutamine through glutaminolysis.

MiRNA-19a-3p has been identified as a suppressor of invasion and metastasis in PC by inhibiting SRY-related high-mobility group box 4 (SOX4), a factor involved in the development, differentiation of cells and organs, as well as the initiation and progression of cancer [73]. In our study, a significant downregulation of serum exosomal miR-19a-3p expression was observed in mCRPC patients. Previous research demonstrated that overexpression of miR-19a-3p led to the downregulation of proteins associated with invasion and metastasis in PC DU145 cells [74]. They reported a significant reduction in miR-19a-3p expression in 121 archived PC tissues, including 76 non-bone metastatic PC tissues and 45 bone metastatic PC tissues. Furthermore, they found that upregulation of miR-19a-3p repressed osteolytic bone lesions. We hypothesize that the downregulation of miR-19a-3p could potentially promote osteolysis and the release of citrate from the bone into the bloodstream.

In an animal model system, an inverse correlation between miR-19a-3p and a key regulatory enzyme of the TCA cycle, citrate synthase, was demonstrated. This study further affirmed that miR-19a-3p potentially interacts with citrate synthase in regions outside the 3′UTR to modulate its mRNA expression [75]. The present study unveiled a negative correlation between miR-19a-3p and serum citrate levels, likely attributable to its impact on citrate synthase expression.

MiR-21 exhibits a dual nature, acting both as an oncogene and a tumor-suppressor [76]. Primarily, miR-21 downregulates phosphatase and tensin homolog (PTEN) expression, fostering the activation of the phosphoinositide-3-kinase–protein kinase B (PI3K/Akt) signaling pathway, thereby propelling cancer progression. Its overexpression impedes apoptosis and plays a crucial role in initiating pro-survival autophagy [77]. MiR-21 significantly contributes to metabolic reprogramming by inducing glycolysis and lactate production, consequently enhancing tumor advancement [78,79]. Moreover, it has been shown that miR-21 plays an essential role in lipid synthesis, fatty acid oxidation, and lipoprotein formation [80,81]. Kanagasabai et al. (2022) documented that miR-21 inactivation resulted in decreased levels of sterol regulatory element-binding protein-1 (SREBP-1), FASN, and acetyl-CoA carboxylase (ACC) in human PC cells. This effect was attributed to the downregulation of insulin receptor substrate 1 (IRS1)-mediated transcription and the induction of cellular senescence [82]. To our knowledge, this is likely the first study reporting the correlation of miR-21 levels with citrate and acetate levels in the serum of mCRPC patients, potentially affecting lipid homeostasis in PC cells. We hypothesize that a reduction in miR-21 expression may partially decrease fatty acid and sterol synthesis, potentially leading to the preferential utilization of acetate over citrate, which could be reflected in their respective serum levels.

Moreover, miR-21 stimulates epithelial–mesenchymal transition (EMT) and upregulates the expression of matrix metalloproteinase-2 (MMP-2) and matrix metalloproteinase-9 (MMP-9), promoting tumor metastasis. MiR-21 is a target of anti-cancer agents like curcumin and curcumol, and its downregulation blocks tumor progression. However, upregulation of miR-21 can result in cancer resistance to chemotherapy and radiotherapy [83]. Variations exist in the findings of these studies, and further investigations are required to determine the suitability of miR-21 as a reliable marker for PC. In line with our study, Damodaran et al. (2021) revealed a significant downregulation of miR-21 in PC patients [84]. Conversely, Kim et al. (2021) reported no significant alterations in miR-21 and miR-141 levels in extracellular vesicles of PC patients compared to the control group [68]. Our investigation shows a decreased expression of serum exosomal miR-21 levels in mCRPC patients. Jokovic et al. (2018) compared plasma and exosomal levels of miRs, indicating elevated exosomal miR-21 levels in PC patients with increased serum PSA values and those with aggressive PC, while plasma samples did not yield significant results [85]. Consequently, their observations suggest a potential prognostic significance for exosomal miR-21 expression levels in PC.

Several studies have observed dysregulated expression of miR-141-3p in various tumor tissues compared to normal controls [86,87]. However, the significance of miR-141-3p in tumor cell proliferation in PC remains unclear to date. MiR-141-3p has garnered considerable attention in cancer research, with its downregulation frequently implicated in the progression and metastasis of various human cancer types. MiR-141-3p inhibits the activation of nuclear factor kappa B (NF-κB) signaling by directly targeting tumor necrosis factor receptor-associated factor 5 and 6, consequently suppressing the invasion, migration, and bone metastasis of PC cells [88]. In animal experiments, it was found that exosomal miR-141-3p exhibited specificity for bone targeting and stimulated osteoblast activity. Specifically, mice administered with miR-141-3p-mimicking exosomes demonstrated evident osteoblastic bone metastasis. Derived from MDA PC 2b cells, exosomal miR-141-3p was identified as a key promoter of osteoblast activity and as a regulator of the bone metastatic microenvironment [89]. Akalin et al.’s (2022) study suggests a potential diagnostic value for miR-141-3p in identifying aggressive PC [90]. Our findings reported no significant alteration in serum exosomal miR-141-3p expression in mCRPC patients.

## 4. Materials and Methods

### 4.1. Study Population

This study received approval from the Ethical Board of Jessenius Faculty of Medicine, Comenius University, and was conducted in accordance with the Declaration of Helsinki. All patients were treated at the Clinic of Urology. Prior to obtaining blood samples, informed written consent was obtained from all participants. The study included 51 patients with mCRPC and 48 patients diagnosed with BPH. Three mCRPC patients were excluded from the study due to low measured PSA levels. Men diagnosed with BPH, aged 55+ and sampled in parallel with mCRPC patient visits, served as the disease control group. BPH diagnosis was confirmed by biopsy at the time of blood sampling.

Examinations of mCRPC patients were conducted at the Clinic of Nuclear Medicine at the University Hospital Martin and Jessenius Faculty of Medicine, Comenius University. Skeletal scintigraphy was performed on every patient, revealing positive findings indicative of confirming multiple or multiple high-volume skeletal metastases in the vast majority of the patients [91]. Most of the scintigraphy examinations were requested by the oncourologist to evaluate the bone metastatic burden before or after the systemic mCRPC treatment, predominantly second or third-line by RaCl_2_. M1a and M1c staging was independently confirmed in several patients by CT, MRI or PSMA PET/CT imaging performed prior to skeletal scintigraphy and/or RaCl_2_ treatment based on previous disease management requirements.

In addition to bone scintigraphy, recent clinical staging and tumor grading information, along with the assessment of PSA levels (Beckman Coulter Access^®^ Hybritech^®^ assay, Beckman Coulter, Fullerton, CA, USA), were carried out during the same day. Venous blood samples were collected in collaboration with the Clinic of Nuclear Medicine and the Clinic of Urology at the University Hospital Martin, Slovakia. Table 4 provides information on the age and PSA levels within both studied groups, as well as the staging characterization and histopathological grading of mCRPC patients.

### 4.2. ^1^H-NMR Metabolomics

Blood handling for metabolomic measurements. Blood specimens were collected using VACUETTE^®^ serum tubes and promptly subjected to centrifugation within one hour of collection. The centrifugation process at 2000 rpm/4 °C/10 min resulted in the separation of serum. The samples were stored at −80 °C until use. Serum denaturation was carried out according to Nagana et al. (2015): 600 µL of methanol was added to 300 µL of blood serum, and the mixture was vortexed and frozen at −24 °C for 30 min. After subsequent centrifugation at 14,000 rpm/room temperature/30 min, 700 µL of supernatant was taken, dried, and stored at −80 °C [92]. Before NMR measurement, the dried matter was soluted in 500 µL of deuterated water and 100 µL of stock solution (phosphate buffer 200 mM, pH 7.4), 0.30 mM TSP-d_4_ (trimethylsilylpropionic acid-d_4_) as a chemical shift reference in deuterated water. Finally, the final solution was transferred into a 5 mm NMR tube.

Data acquisition and processing. Nuclear magnetic resonance (NMR) data were acquired using a 600 MHz NMR spectrometer, Avance III, from Bruker, equipped with a TCI CryoProbe (Bruker, Bremen, Germany), operating at a temperature of 310 K. Initial settings, including field shimming, receiver gain, and water suppression frequency, were optimized using an independent sample and then applied consistently across all measurements. After preparation, samples were stored in a Sample Jet automatic machine for a maximum of 2 h, kept at approximately 5 °C. Before measurement, each sample underwent preheating at 310 K for 5 min. An exponential noise filter with 0.3 Hz line broadening was applied prior to a Fourier transform. All data were subjected to zero-filling. The acquisition order of samples was randomized for the experiments.

Bruker profiling protocols were modified as follows: NOESY with presaturation (noesygppr1d): FID size 64k, dummy scans 4, number of scans 64, spectral width 20.4750 ppm; profiling cpmg (cpmgpr1d, L4 = 126, d20 = 3 ms): number of scans 256, spectral width 20.4750 ppm. For a subset of 15 randomly chosen samples, 2D spectra were measured: cosy with presaturation (cosygpprqf): FID size 4 k, dummy scans 8, number of scans 16, spectral width 16.0125 ppm. Homonuclear J-resolved (jresgpprqf) utilized a FID size of 8 k, 16 dummy scans, 32 scans. All experiments were conducted with a relaxation delay of 4 s. Across all samples, the half-width of the TSP-d_4_ signal was consistently maintained below 1.0 Hz.

Spectra were solved using multiple references, including a human metabolomic database (www.hmda.ca) [93], the Chenomx v7 software, an internal metabolite database, and a review of relevant metabolomic literature [92]. Proton chemical shifts were compared with the TSP-d_4_ signal assigned a chemical shift of 0.000 ppm. Peak multiplicities were verified through J-resolved spectra, while homonuclear cross-peaks were confirmed using 2D cosy spectra.

All spectra were binned into bins of 0.001 ppm in size. No additional normalization methods were applied to the data since an equal volume of blood serum was used for all measurements. Subsequently, the intensities of selected bins were summed exclusively for spectra subregions where only one metabolite was assigned or minimally affected by other co-metabolites. Metabolites exhibiting weak intense peaks or significant peak overlap were omitted from the analysis. The resulting values were utilized as relative concentrations of metabolites in each sample.

### 4.3. Exosomal miRs Expression

The serum exosomal miR-15a, miR-16, miR-19-3p, miR-21 and miR-141a-3p were selected based on their demonstrated high diagnostic efficacy in previous studies. Total exosomal RNA containing miR was isolated through a spin column-based method (Serum/Plasma ExoRNAse Midi Kit, Qiagen, Hilden, Germany) following the manufacturer’s instructions. Before isolation of total RNA, 0.5 mL of serum was initially centrifuged at 3000× *g*/room temperature/15 min. The RNA purity and quantity were measured using a NanoPhotometer (Implen, GmbH, Germany).

Total exosomal RNA was reverse transcribed into cDNA using a TaqMan MicroRNA Reverse Transkription Kit (Applied Biosystems, Foster City, CA, USA), and TaqMan MicroRNA Assay Kit (catalog no. 4427975; Applied Biosystems) containing miR RT primers of selected miRs: miR-15a (assay ID: 000389), miR-16(assay ID: 000391), miR-19a-3p (assay ID: 000395), miR-21 (assay ID: 000397), miR-141a-3p (assay ID: 000463), and miR-1233 (assay ID: 002768). QRT-PCR was performed using a TaqMan Universal PCR Master Mix II, no UNG (Applied Biosystems) and carried out by CFX96 Real-Time PCR Detection System (Bio-Rad Laboratories, Hercules, CA, USA) and processed as described previously [94,95]. The expression levels of exosomal miRs were normalized to miR-1233, identified as the most suitable control for data normalization. Each sample was analyzed in duplicate, and the data were processed using the 2^−ΔΔCt^ method.

### 4.4. Statistical Analysis

^1^H-NMR metabolomics. The distribution of metabolic data was approached by a Shapiro–Wilk test, which is suitable for detecting non-normality in a small sample size (*n* < 50) [96]. The test rejected normality in about half of the data. The null hypothesis of equality of population medians among mCRPC patients and BPH patients was established by a non-parametrical Mann–Whitney U test. A *p*-value of 0.05 was used as a threshold to claim significance. Calculations were carried out in OriginPro 2019 (v.9.6.0.172, OriginLab, Northampton, MA, USA) and Matlab (v. 2015b, Mathworks, Natick, MA, USA).

Exosomal miR expression. We employed the Shapiro–Wilk test to assess the normality of the dataset. Subsequently, we utilized the non-parametric Mann–Whitney U test to discern significant differences between groups. Differences were deemed statistically significant at a threshold of *p* < 0.05. Visualization of the results was accomplished using Origin Pro 2019 (academic licence).

## 5. Conclusions

In conclusion, this study identified previously unreported metabolomic alterations in lactate, citrate, and acetate in conjunction with serum exosomal miR-15a, miR-16, miR-19a-3p, and miR-21, which may have the potential to distinguish mCRPC from BPH. Nonetheless, several limitations were identified. Firstly, the relatively small sample size restricts the clinical applicability of our findings, necessitating larger-scale investigations in the future. Secondly, the utilization of BPH patients as the control group raises considerations. BPH is intricately linked with age, affecting approximately 50% of men over 50 years and over 80% of men over 80 years [97]. Consequently, we opted to employ the BPH group as the control, given the challenge of selecting age-matched controls without BPH in this demographic. Despite these limitations, we posit that our study contributes to a deeper comprehension of mCRPC tumor biology. In the future, larger studies should be conducted using samples of healthy control, BPH, PC and mCRPC patients for crucial insights into the diagnosis and classification of PC.

## Figures and Tables

**Figure 1 ijms-25-02630-f001:**
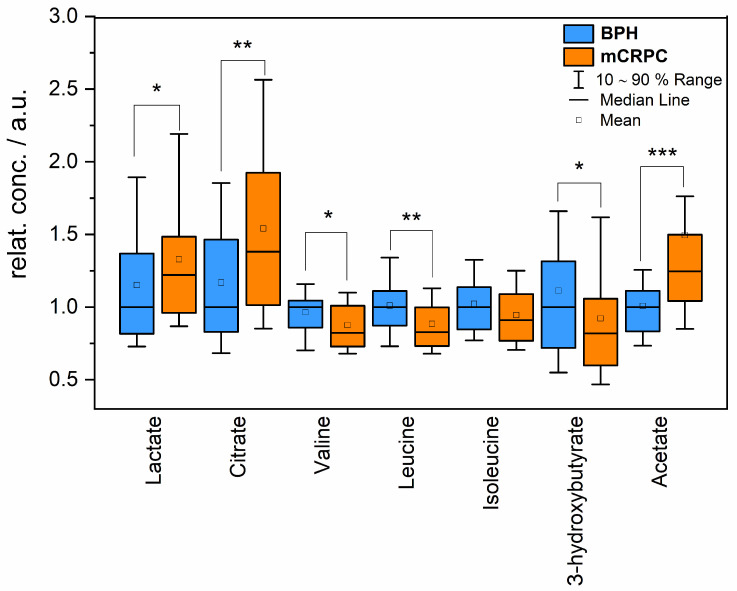
Relative concentrations of circulating metabolites in mCRPC patients (orange) and BPH patients (blue). Data were normalized to the median of the BPH set to 1, * *p* < 0.05; ** *p* = 0.05–0.01; *** *p* < 0.001.

**Table 1 ijms-25-02630-t001:** Statistical comparison of relative concentrations of blood serum metabolites in mCRPC patients and BPH group. *p* value derived from Mann–Whitney U test, percentual change derived from medians. Positive values indicate heightened levels in mCRCP patients compared to BPH, whereas negative values indicate lowered levels.

Metabolite	Percentage Change mCRPC vs. BPH Group	*p*-Value
Lactate	22%	0.04
Citrate	38%	0.003
Valine	−18%	0.02
Leucine	−18%	0.002
Isoleucine	−11%	0.06
3-hydroxybutyrate	−19%	0.03
Acetate	24%	0.00002

**Table 2 ijms-25-02630-t002:** Fold regulation levels of exosomal miRs (miR-15a, miR-16, miR-19a-3p, miR-21, and miR-141a-3p) between mCRPC and BPH patients.

miR	Fold Regulation +/−	*p*-Value
miR-15a	−2.00	0.006
miR-16	−3.24	4.33 × 10^−6^
miR-19a-3p	−2.78	1.11 × 10^−5^
miR-21	−2.39	0.003
miR-141a-3p	+1.23	0.41

+, upregulation (comparing to BPH group); −, downregulation (comparing to BPH group).

**Table 3 ijms-25-02630-t003:** Correlation analysis of serum exosomal miRs and metabolites.

		miR-15a	miR-16	miR-19a-3p	miR-21	miR-141a-3p
Lactate	Pearson Corr.	0.35	0.31	0.33	0.27	0.01
	*p*-value	0.01	0.03	0.03	0.06	0.92
Citrate	Pearson Corr.	−0.47	−0.24	−0.35	−0.47	−0.24
	*p*-value	9.23 × 10^−4^	0.11	0.02	7.61 × 10^−4^	0.09
Valine	Pearson Corr.	0.05	0.05	0.15	0.14	−0.09
	*p*-value	0.73	0.071	0.33	0.35	0.55
Leucine	Pearson Corr.	−0.04	−0.02	0.05	0.08	−0.09
	*p*-value	0.78	0.89	0.72	0.55	0.55
Isoleucine	Pearson Corr.	−0.23	−0.09	−0.07	−0.12	−0.08
	*p*-value	0.11	0.52	0.62	0.44	0.59
3-hydroxybutyrate	Pearson Corr.	0.28	0.09	0.17	0.33	0.05
	*p*-value	0.05	0.51	0.24	0.03	0.72
Acetate	Pearson Corr.	0.16	0.01	0.30	0.46	0.42
	*p*-value	0.29	0.96	0.04	0.001	0.003
Glutamine	Pearson Corr.	0.28	0.28	0.21	0.19	−0.14
	*p*-value	0.05	0.05	0.16	0.21	0.36
Lysine	Pearson Corr.	−0.44	−0.22	−0.29	−0.39	−0.04
	*p*-value	0.002	0.14	0.05	0.006	0.81

**Table 4 ijms-25-02630-t004:** Description of the cohort under study.

	mCRPC Patients	BPH Patients	*p*-Value
Number	51	48	
Age (years, mean ± SD)	73.5 ± 7.65	67.5 ± 6.54	<0.05
Range	(58–85)	(55–84)	
PSA (ng/mL, mean ± SD)	103.5 ± 1478.7	3.29 ± 9.77	<0.05
Range	(2.2–9506)	(0.2–66.5)	
T Staging	No.	N/A	–
T2	2		
T3	35		
T4	14		
N staging	No.	N/A	–
N0	40		
N1	11		
M staging *	No.	N/A	–
M1a	8		
M1b	51		
M1c	4		
Gleason score	No.	N/A	–
6	1		
7	9		
8	17		
9	20		
10	4		

* one or more staging for each patient. N/A, not applicable.

## Data Availability

The data presented in this study are available on request from the corresponding author. The data are not publicly available due to the nature of the research and sensitivity of information obtained from the research subjects.

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
