# Peer review of "Integrative Metabolomic Analysis of Serum and Selected Serum Exosomal microRNA in Metastatic Castration-Resistant Prostate Cancer"

_ijms, 2024, doi:10.3390/ijms25052630_

Round 1

Reviewer 1 Report

Comments and Suggestions for Authors

Dear Authors, 

I reviewed with interest the paper entitled “Integrative metabolomic analysis of serum and selected serum exosomal microRNA in metastatic castration-resistant prostate cancer”.

First, I would strongly congratulate with the authors for their work for this study, which covers an interesting topic such as evaluating potential biomarkers for mCRPC. 

I found the present study interesting - no concerns with English language editing.

- The title is clear and descriptive of what authors have explored in their work.

- The Introduction is clear, fluent to read, and provides a background which is relevant to the study. However, I would suggest adding main tools which are not mentioned at all, yet are pivotal for PCa diagnosis and management, such as biomarkers (e.g., DOI: 10.1097/JU.0000000000001361; doi: 10.1111/iju.13734) and PSAD (e.g., DOI: 10.3389/fonc.2021.693684). As minor comment, I would suggest a proper use of acronym (e.g., PC for prostate cancer); some of them are not always used while some others were not spelled out when used for the first time (e.g., lines 40, 48, 89, 185..). The aim is properly provided at the end of the Introduction section. 

- My main concerns are on Materials and Methods section and, more in general, study design. More specifically, the comparison was made between BPH patients (“healthy controls”, “benign disease”) and mCRPC patients (advanced metastatic disease)? The evaluation should be done also with PC patients which are not castration resistant or metastatic to give evidence on castration resistant or metastatic disease. Otherwise, is it not possible to draw conclusions on these 2 conditions. Moreover, how was BPH diagnosis done? By biopsies? Could you exclude that these patients have also PC? Why all patients had scintigraphy, yet whole-body CT scan was not mentioned? Was that performed for systemic evaluation? In the light of that, Discussion with implementation of limitations of the study, and Conclusions should be modified.

I have not further suggestions.

Author Response

Dear Reviewer,

We would like to express our gratitude for your letter and the constructive comments provided regarding our manuscript. We have thoroughly reviewed your suggestions and have implemented the necessary corrections accordingly. All modifications and edits have been highlighted in color within the revised version of the manuscript.

We trust that these revisions address the issues raised and improve the quality and clarity of our manuscript. Thank you once again for your valuable feedback and for the opportunity to enhance our work.

Sincerely,

Sivoňová Kmeťová Monika

Reviewer 2 Report

Comments and Suggestions for Authors

The study “Integrative metabolomic analysis of serum and selected serum exosomal microRNA in metastatic castration-resistant prostate cancer” is mainly descriptive. This study has a certain value in accumulating knowledge about such a socially significant and practically incurable disease as metastatic castration-resistant prostate cancer. However, the study has several disadvantages.

Major comments:

1. Why were sera from healthy donors not taken as a control? The study would have been more complete if the authors had compared three groups: healthy donors, mCRPC patients, and BPH patients. Blood serum was used as speciemen; obtaining serum from healthy volunteers should not be as difficult as obtaining tissue samples, for example.

In order to write about the diagnostic significance of markers, it is necessary to compare the levels of these markers in patients with mCRPC and healthy people. This article compares marker levels in patients with mCRPC and benign prostatic hyperplasia, which can only be used for differential diagnosis of these two diseases.

2. Figure 1. The presented figure is not a boxplot, but a bar chart. The results can be presented like this, but then the text must be changed (line 101). However, it is better to modify the figure and present it as a boxplot, which allows the reader to see the mean (median), first and third quartiles, and spread of data in each group being compared.

3. Analysis of the correlation of exosomal microRNA expression (lines 150-156 and Table 3) does not carry any meaning. This part of the result may be removed without affecting the article. Alternatively, the Discussion should show the significance of the calculated correlations. Also in the Discussion, more emphasis should be placed on the role of the studied microRNAs in energy, lipid metabolism, etc., which is confirmed or refuted by the results presented in Table 4.

4. In the Discussion it is written about the role of miR-141, however, miR-141a-3p was studied (see Materials and Methods). The authors should edit the Discussion so that it describes the biological role of miR-141a-3p specifically (not miR-141-5p) and perhaps indicate the difference or lack thereof between miR-141a-3p and miR-141-3p. This is important because the authors write that the choice of microRNAs to investigate was based on previously published data.

Minor comments:

1. Indicate the AssayIDs for the used MicroRNA Assay Kits.

2. Table 1. Need clarification of what the “-” before the number in the table means. Please provide explanations similar to Table 2.

3. Table 5. What does “P-value” mean in the “Number” line? The significance/reliability of what was calculated?

Author Response

(The authors gave the same response as above.)

Round 2

Reviewer 2 Report

Comments and Suggestions for Authors

I thank the authors for their work in revising the article.

The authors gave a detailed answer to the question about the reason for choosing the comparison groups (control/experiment), made appropriate changes to the text of the article, changed Figure 1 in accordance with the comment, and changed the tables.

All the authors' answers are accepted, except for the answer to the comment about AssayIDs for the used MicroRNA Assay Kit. Please indicate exactly AssayIDs.

  MicroRNA Assay Kit (Catalog no. 4427975) is a kit consisting of two tubes, one containing primers for reverse transcription, and the other containing primers and probes for PCR. The catalog number is the same for all microRNAs, regardless of miRBase ID or Assay Name. When ordering, not only the catalog number is indicated, but also the AssayID, which specifies the primers for which exact microRNA are ordered. Recently, articles have appeared comparing the properties of microRNAs derived from the -3p and -5p arms of the precursor (pre-microRNA), so it is necessary to indicate the AssayID. As an example, I am attaching screenshots from the Applied Biosystems (ThermoFisher Scientific) website where AssayIDs for miR-15a-3p and miR-15a-5p are shown. Please note that the AssayID for miR-15a* (hsa-miR-15a-3p) and miR-15a (hsa-miR-15a-5p) are different, but the catalog number is the same.

Line 342. Typo in the word ‘protein’.

Author Response

Dear Reviewer,

We would like to express our gratitude for your letter and the constructive comments provided regarding our revised manuscript. We have thoroughly reviewed your suggestions according miRNA and have implemented the necessary corrections accordingly. All modifications and edits have been highlighted in color within the revised version of the manuscript.
